# River channel connectivity shifts metabolite composition and dissolved organic matter chemistry

Laurel M. Lynch [1,2], Nicholas A. Sutfin [3], Timothy S. Fegel[4], Claudia M. Boot [2,5], Timothy P. Covino [2,6] & Matthew D. Wallenstein [2,6]

Biogeochemical processing of dissolved organic matter (DOM) in headwater rivers regulates aquatic food web dynamics, water quality, and carbon storage. Although headwater rivers are critical sources of energy to downstream ecosystems, underlying mechanisms structuring DOM composition and reactivity are not well quantified. By pairing mass spectrometry and fluorescence spectroscopy, here we show that hydrology and river geomorphology interactively shape molecular patterns in DOM composition. River segments with a single channel flowing across the valley bottom export DOM with a similar chemical profile through time. In contrast, segments with multiple channels of flow store large volumes of water during peak flows, which they release downstream throughout the summer. As flows subside, losses of lateral floodplain connectivity significantly increase the heterogeneity of DOM exported downstream. By linking geomorphologic landscape-scale processes with microbial metabolism, we show DOM heterogeneity increases as a function of fluvial complexity, with implications for ecosystem function and watershed management.

[1] Section of Soil and Crop Sciences, College of Agriculture and Life Sciences, Cornell University, Ithaca, NY, USA. [2] Natural Resource Ecology Laboratory, Colorado State University, Fort Collins, CO, USA. [3] Department of Earth, Environmental, and Planetary Sciences, Case Western Reserve University, Cleveland, OH, USA. [4] United States Forest Service, Rocky Mountain Research Station, Fort Collins, CO, USA. [5] Central Instrument Facility, Department of Chemistry, Colorado State University, Fort Collins, CO, USA. [6] Department of Soil and Crop Sciences, Colorado State University, Fort Collins, CO, USA. Correspondence and requests for materials should be addressed to L.M.L. (email: lml262@cornell.edu)

Dissolved organic matter (DOM) accounts for the vast majority of reactive carbon flowing through aquatic ecosystems[1,2]. Given the central role of DOM in global carbon cycling, nutrient export, and food web dynamics[3,4], unraveling processes controlling the flux and transformation of DOM in watersheds is essential to understanding potential changes resulting from land use and climate change. Headwater rivers exert disproportionate control on fluvial biogeochemistry[5] by setting initial conditions that cascade to the entire watershed[4] and by providing a critical source of chemical energy to downstream ecosystems. However, current state of the sciences lacks a quantitative understanding of the relationship between seasonal hydrology and geomorphic complexity in structuring DOM composition and reactivity.

In mountainous watersheds, the majority of DOM enters the fluvial network in small headwater tributaries[6]. By some estimates, half of this carbon is mineralized through microbial and photochemical degradation[7], while the fraction remaining within the watershed is retained on floodplains, exported further downstream, or generated in situ through autochthonous production or heterotrophic fractionation[2,8,9]. River geomorphic complexity—the physical heterogeneity in river channel geometry and planform[10]—is a critical factor controlling the residence time and concentration of DOM residing within fluvial networks[11,12]. This is particularly evident in mountainous channel networks, where rivers alternate between segments with multiple channels of flow (complex or multi-thread) and segments with a single channel of flow (simple) across the valley bottom[13]. Here we show that channel complexity is a significant driver of DOM composition, which varies as a function of lateral river-floodplain hydrologic connectivity.

Stable multi-thread channels develop in broad, relatively unconfined valleys of Rocky Mountain National Park, where channel-spanning logjams and beaver (*Castor canadensis*) activity dissipate transport energy and create sites of flow diversion[13,14]. These multi-thread channels serve as retention zones that enhance the transient storage of DOM[15] and may increase geophysical opportunities for microbial metabolism[3]. Flume studies have shown that microbial diversity and uptake of chemically complex DOM is positively correlated with physical complexity[16,17]. Diversification of metabolic pathways in complex channels may thus translate into more extensive DOM processing prior to downstream export. In contrast, single-thread channel segments commonly occur in steep-gradient valleys and are characterized by high transport capacities and limited sediment, water, and fluvial carbon storage[9].

Many once complex, multi-thread segments of wide valleys have been transformed to simple, single-thread segments resulting from flow regulation, land-use changes, and beaver removal[9]. Ensuing loss of geomorphic complexity results in rapid export of DOM through the landscape, potentially minimizing microbial transformation of DOM and limiting river-floodplain hydrologic connectivity, and floodplain carbon storage[9]. While geomorphic complexity appears to be an important driver of DOM composition, limited analytical resolution has precluded detailed investigation[18]. As a result, it remains unclear whether decomposition results in the convergence of DOM molecular composition toward a core set of metabolites, or persistence of the original, chemically diverse molecular profile[19]. Resolution of this debate is essential in determining how DOM is cycled within a landscape.

Strong fluctuations in river flows, resulting from seasonal snowmelt or stormflow events, may interact with geomorphic complexity to control molecular patterns of DOM composition. Elucidating the influence of environmental drivers on DOM composition remains challenging because of tremendous chemical heterogeneity[18] and the complexity of underlying spatiotemporal mechanisms influencing its composition. Mass spectrometry has vastly improved our understanding of the composition and cycling of DOM compounds varying in physicochemical reactivity. While fluorescence, nuclear magnetic resonance, pyrolysis gas chromatography mass spectrometry (GC-MS), and Fourier transform ion cyclotron resonance mass spectrometry (FT-ICR-MS) approaches are routinely used to probe DOM composition, they group DOM complexity into broad functional groups or compound classes. In contrast, electron ionization GC-MS (EI GC-MS) can identify individual metabolites from complex mixtures of environmental DOM[20]. This detailed characterization of DOM allows inferences about the underlying patterns of microbial metabolism that structure the reactivity and fate of DOM flowing throughout the fluvial network.

In this study, we collected surface and hyporheic water samples to monitor changes in fluvial chemistry through space (spanning 3 km of valley length in two watersheds with river networks of differing geomorphic complexity) and time (eight time points from high to low river flows spanning May to October) (Table 1 and Supplementary Figure 1). We selected two river segments (simple network and complex network), with nearly identical climate regimes and underlying geology that differ substantially in channel complexity. Three sites within the simple network were sampled. Seven sites within the complex network were sampled, including: a beaver pond that has limited surface water

**Table 1 Physical characteristics of sampling sites**

|  | Simple | Upper | Meadow complex | Lower | Pond |
|---|---|---|---|---|---|
| Watershed | UBM | NSV | NSV | NSV | NSV |
| Channel planform | Single-thread | Single-thread | Multi-thread | Single-thread | Pond |
| Lateral river-floodplain connectivity | Low | Low | High | Low | None |
| Longitudinal connectivity | Continuous | Continuous | Intermittent | Continuous | None |
| Number of sites | 3 | 2 | 2 | 2 | 1 |
| Watershed area (m) | 21.6 | 88.9 | 88.9 | 88.9 | 88.9 |
| Channel width (m) | 0.71 | 12.2 | 14.5 | 20 | 12.8 |
| Valley width (m) | 163.9 | 59.6 | 246.8 | 27.5 | 246.9 |
| Confinement | 0.004 | 0.204 | 0.059 | 0.725 | 0.052 |
| Elevation (m) | 2,572.51 | 2,561.08 | 2,544.33 | 2,519.76 | 2,545.85 |

These are located in UBM (simple network) and NSV (complex network) rivers, Rocky Mountain National Park, Colorado. Channels with a confinement value < 0.2 are considered unconfined while those with a confinement value ≥ 0.2 are considered confined. Confined is defined as the ratio of the channel width versus the valley width
*UBM* Upper Beaver Meadow, *NSV* North Saint Vrain

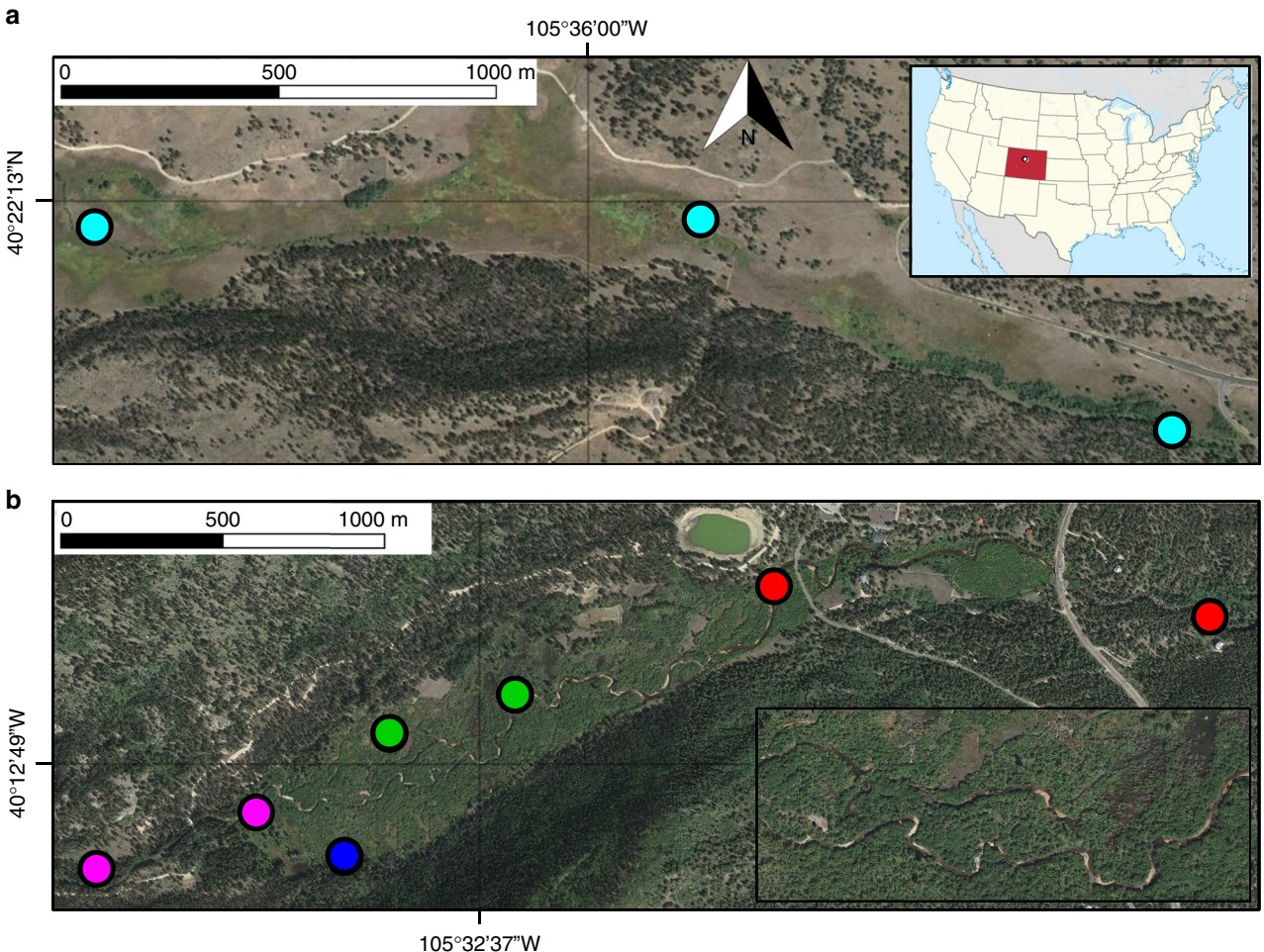

**Fig. 1** Site map of the simple and complex river channel networks sampled for this study. **a** From the simple network (Upper Beaver Meadows; UBM), we present data from three single-channel sites (light blue symbols). **b** From the complex network (North Saint Vrain Creek; NSV), we present data from two main channel sites in the upper confined segment (pink symbols), two side-channel features (green symbols), and a disconnected beaver pond (blue symbol) in the active beaver meadow complex, and two main channel sites in the lower confined segment (red symbols). Symbol colors match Fig. 3. The headwaters of UBM and NSV are located in Rocky Mountain National Park, Colorado, USA. Map data ©2018 Google

connectivity to the main channel, and confined upstream and downstream (simple morphology) river segments that bracket the main stem and side-channel features within a wide-valley, multi-thread beaver meadow complex (Fig. 1). We used complementary approaches, including EI GC-MS and fluorescence spectroscopy (excitation-emission matrices), to quantify spatial and temporal differences in DOM molecular composition. We also determined the potential role of geomorphic complexity, hydrologic connectivity, and river discharge in structuring DOM molecular composition. We posit that river segments with high geomorphic complexity increase the chemical diversity of DOM exported downstream, and maximal chemical diversity occurs during high-flow periods and declines over time as flows subside.

## Results

**Influence of channel connectivity on DOM composition.** The variability in DOM chemical composition within the complex river network increased as connectivity with the main channel declined through the summer. In contrast, the chemical composition of DOM exported through the simple network was not influenced by seasonal changes in river flows. Therefore, for each sampling point we used an average of the three sampling locations along the simple network for subsequent statistical analyses.

Bulk carbon concentrations collected from all sampling locations were significantly different across time and space ($p < 0.01$, $F_{8,39} = 3.55$; Table 2). Of all the potential explanatory geomorphic and environmental variables tested, only valley confinement was a significant predictor of DOC concentrations ($p < 0.001$, Supplementary Table 1), which were higher during intermediate flows than peak or low flow conditions. Within the complex network, DOC concentrations were significantly higher in the beaver pond than all other sites ($p < 0.01$) but were not different between upper and lower confined segments (Table 2). We also observed an effect of landscape position on hyporheic carbon and nitrogen concentrations, such that sites with lower transport energy (beaver pond and beaver meadow) had higher carbon and nitrogen concentrations than freely flowing reaches ($F_{8,39} = 2.44$; $p < 0.05$).

Total dissolved nitrogen (TDN) concentrations were twice as high in the beaver pond relative to all other sites ($p < 0.01$); hyporheic nitrogen concentrations decreased across all sites throughout the season ($p < 0.01$). Regressing TDN against discharge revealed nitrogen concentrations were highest during peak river flows ($R^2 = 0.46$). Total C:N varied spatially and temporally ($F_{8,39} = 4.22$; Table 2); relative nitrogen availability was higher in the beaver pond and simple channel than the beaver meadow complex and downstream confined channel

**Table 2 Biogeochemical characteristics of surface water and hyporheic sediments**

| ID | DOC | TDN | NO₃⁻ | NH₄⁺ | Hyporheic DOC | Hyporheic TDN | Shannon-Weiner |
|---|---|---|---|---|---|---|---|
| Landscape position | | | | | | | |
| Simple | 4.00 (0.53) | 0.25 (0.03) | 0.08 (0.02) | 0.07 (0.01) | 4.81 (0.54) | 0.35 (0.05) | 3.45 (0.09) |
| Pond | 9.90 (1.95) | 0.61 (0.15) | 0.03 (0.01) | | 8.10 (1.49) | 0.68 (0.23) | |
| Upper | 2.70 (0.55) | 0.33 (0.05) | 0.11 (0.01) | | 3.41 (0.65) | 0.58 (0.17) | |
| Meadow | 4.27 (0.49) | 0.39 (0.05) | 0.08 (0.02) | | 8.64 (2.29) | 1.20 (0.29) | |
| Lower | 3.42 (0.71) | 0.31 (0.04) | 0.09 (0.02) | | 4.41 (1.02) | 0.73 (0.21) | |
| Sampling period | | | | | | | |
| 1: May 1 | 6.49 (0.66) | 0.49 (0.07) | 0.11 (0.01) | 0.17 (0.06) | 6.20 (1.76) | 2.13 (0.20) | 3.45 (0.12) |
| 2: May 14 | 5.17 (0.67) | 0.35 (0.02) | 0.08 (0.02) | 0.08 (0.01) | | 1.28 (0.47) | |
| 3: Jun 8 | 4.72 (0.29) | 0.47 (0.04) | 0.11 (0.03) | 0.08 (0.01) | | 0.54 (0.06) | |
| 4: Jun 30 | 3.58 (0.49) | 0.38 (0.04) | 0.06 (0.02) | 0.08 (0.01) | | 0.49 (0.08) | |
| 5: Jul 21 | 4.32 (1.75) | 0.32 (0.06) | 0.04 (0.01) | 0.06 (0.01) | | 0.54 (0.11) | |
| 6: Aug 17 | 8.22 (3.28) | 0.49 (0.15) | 0.03 (0.01) | 0.05 (0.00) | | 0.37 (0.08) | |
| 7: Sep 21 | 4.57 (2.68) | 0.23 (0.11) | 0.04 (0.01) | 0.05 (0.00) | | 0.42 (0.09) | |
| 8: Oct 28 | 4.64 (2.49) | 0.43 (0.28) | 0.06 (0.01) | 0.05 (0.00) | | 0.31 (0.10) | |
| Source of variance | | | | | | | |
| LP | ** | * | *** | ns | * | ** | ns |
| SP | ** | ns | * | *** | ns | *** | ns |

These are organized by landscape position and sampling period (discharge) at Upper Beaver Meadows (simple) and North Saint Vrain (pond, upper, meadow, and lower). The average followed by the standard error (±1 S.E.) in parentheses for dissolved C, N (DOC, TDN), inorganic N (NO₃⁻, NH₄⁺), and a metabolite chemodiversity index (Shannon-Weiner). Aside from the unitless chemodiversity index, all values are reported in mg/L. The level of significance from the two-way analysis of variance model for samples arranged by LP and SP are reported as *$p < 0.05$, **$p < 0.01$, ***$p < 0.001$, or nonsignificant (ns). For ns comparisons, a master mean, integrated across landscape position or sampling period, is provided. Specific comparisons were explored using Tukey HSD $t$-tests with Satterthwaite-Welsch approximations for degrees of freedom (reported in the Results section)

*DOC* dissolved organic carbon, *TDN* total dissolved nitrogen, *LP* landscape position, *SP* sampling period

($p < 0.05$). The fluorescence characteristics of ultraviolet-visible (UV-VIS) DOM (FDOM) exhibited strong temporal dynamics but were not influenced by geomorphic complexity (Supplementary Table 2). We observed the lowest contribution of simple aromatic proteins (regions I and II) and soluble microbial-type proteins (region IV, also interpreted as microbial exudates) during peak river flows, with the contribution of these regions to total FDOM doubling throughout the summer (region I: $F_{8,39} = 2.84$; Fig. 2a). In contrast, the contribution of fulvic (region III)- and humic (region V)-type acids to FDOM decreased in relative intensity by 60% throughout the season, except for a highly enriched signal exported during peak discharge (region III: $F_{8,39} = 2.84$; Fig. 2b). Regressing region I against region III revealed a strong, negative correlation ($R^2 = 0.89$), suggesting high levels of autochthonous productivity occur during periods of low allochthonous subsidy (Supplementary Figure 2). Discharge alone was a significant predictor of regions I ($p < 0.01$, $R^2 = 0.27$) and III ($p < 0.05$, $R^2 = 0.22$), supporting the control of seasonality on FDOM composition (Supplementary Table 1).

**Metabolite distribution**. We observed a total of 2472 mass spectral features using untargeted EI GC-MS. We resolved these features using cluster analysis, resulting in 259 unique compounds that each consisted of between 3 and 57 mass spectral fragments (Supplementary Figure 3)[21]. We were able to annotate and classify 10% of these compounds[22–24], which primarily consisted of sugars, organic acids, lipids, and lignin-derived aromatics (Table 3). Due to limitations in current mass spectral libraries we were not able to annotate the majority of compounds driving sample separation across geomorphic complexity and sampling period. However, we were able to use changes in metabolite composition to separate samples in ordination space.

Over half of all profiled metabolites (both annotated and not) exhibited significant separation across regions of varying geomorphic complexity (Supplementary Figure 4a). Combined, partial least squares discriminant analysis (PLS-DA) components 1 and 2 explained 43% of the variance, and retention of four axes maximized classification performance ($R^2 = 0.69$; coefficient of

variation (CV) accuracy = 0.56). As PLS-DA classification can be susceptible to overfitting, we used $Q^2$ scores to estimate the predictive ability of the model through cross-validation[25]. While our strongly positive score ($Q^2 = 0.43$) suggests the model is not overfitted[26] we also include results from two unsupervised approaches (principal components analysis (PCA), Supplementary Figure 5, and random forest, Supplementary Figure 6) that display similar trends. The five primary VIP compounds, or metabolites that contributed most to sample separation across geomorphic complexity, had slightly lower parent ion $m/z$ values (352.0) than average (402.2), but low annotation scores preclude meaningful metabolite assignments (Supplementary Table 3).

Spatial variability in metabolite composition among upstream and downstream reaches was distinctly different between the simple and complex meadow sites. The beaver pond and simple system, which represent end-members of hydrologic (dis)connectivity, had divergent metabolite profiles (Fig. 3a), indicating a strong influence of geomorphic complexity and hydrologic connectivity on metabolite distribution. Within the complex network, the upper and lower confined reaches showed minimal overlap in metabolite distribution, while the beaver meadow complex, situated in physical space between the two sites, shared metabolites with each. The upper confined sites and beaver meadow complex sites were more dissimilar than the lower confined site, which contained metabolites very similar to, and within the range of, the meadow site (Fig. 3d). We observed a similar pattern when considering only annotated metabolites. The five primary VIP compounds contributing to metabolite separation across geomorphic complexity were related to byproducts of decomposition, including nonanoic and dodecanoic acids (medium-chain saturated fatty acids), glyceric acid (a three-carbon sugar acid), and isovanillic acid (a derivate of lignin degradation) (Table 3)[23]. These compounds had the highest scores in the beaver pond and simple network, the lowest scores in the upper confined channel, and relatively low scores within the beaver meadow complex and lower confined channel (Supplementary Figure 7a).

Metabolites also separated across sampling period (Fig. 3b), although only 40% exhibited significant temporal variability

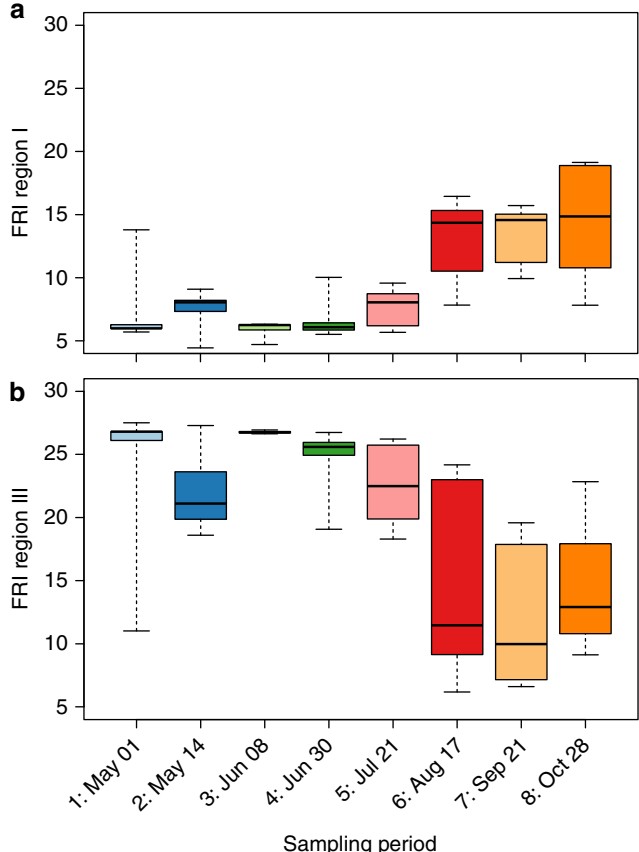

**Fig. 2** Boxplots represent seasonal distributions of excitation-emission matrix spectra (EEMS) regions. EEMS region I: simple aromatic proteins (**a**), and EEMS region III: fulvic-type acids (**b**) across sampling period (*n* = 49) within the complex river network. Regions II and IV (simple aromatic proteins and soluble microbial byproducts) follow similar seasonal patterns as region I, while region III and V (humic-type acids) are related. Each box spans the interquartile range; whiskers extend to the minimum and maximum of the distribution

(Supplementary Figure 4b). Together, the first two PLS-DA components explained 45% of the variance, with maximal classification performance achieved with five axes ($R^2 = 0.96$; CV accuracy = 0.44, $Q^2 = 0.80$). The five primary annotated VIP compounds were typically enriched early in the season and included lumichrome (a flavin pigment), glycerol (a three-carbon sugar acid), 2-hydroxypyridine (potentially involved in DNA synthesis), and several polyethylene glycols (ether compounds) (Table 3 and Supplementary Figure 7)[23].

Samples collected during the rising limb and peak discharge (Fig. 3c) were composed of significantly different metabolites than samples collected during baseflow (Fig. 3d), with typically lower parent ion *m/z* values. Early in the season, samples did not separate by landscape position (*p* = 0.81), suggesting high hydrologic connectivity during high-flow periods homogenizes the DOM pool and exports a core set of metabolites (Fig. 3c). In contrast, we observed significant sample separation across geomorphic complexity during late season sampling (*p* = 0.01), suggesting metabolite profiles diverge during low flow periods (Fig. 3d).

**Environmental drivers of metabolite diversity.** We identified six environmental drivers associated with metabolite composition (Fig. 4) using constrained analysis of principal coordinates[27]. Within ordination space, principal coordinate 1 explained 43% of

variation across the dataset and principal coordinate 2 explained an additional 30%. Dissolved organic carbon (DOC) concentrations emerged as the primary factor influencing metabolite composition and was negatively correlated with discharge (Fig. 4), where high discharge exported metabolites with relatively lower parent ion *m/z* values than low flow periods. Additionally, metabolites clustering with higher DOC concentrations had higher parent ion *m/z* values than those associated with proteins (excitation-emission matrix spectra (EEMS) region I, *p* < 0.003) and snowpack extent (*p* < 0.006), indicating lower chemical reactivity. Samples clustering with proteins were typically collected later in the summer, when low flows reduced hydrologic connectivity across the landscape. These patterns suggest more complex river corridors export more chemically diverse, reactive metabolites from multi-thread channels relative to their single-thread counterparts.

**Discussion**

Our results suggest that fluvial chemistry and the molecular composition of DOM exported through mountainous watersheds are strongly influenced by seasonality and channel geomorphology. Broad indices of DOM composition, including optical fluorescence (UV-VIS) and DOC and TDN concentrations, were strongly related to in-stream flows. While these bulk chemical approaches revealed the influence of seasonality, we found the application of non-targeted metabolomics—a higher-resolution approach capable of profiling both known and unknown metabolites[21,22]—helped resolve the influence of landscape complexity on DOM chemistry. We suggest linking complex environmental drivers to ecosystem metabolism is critical in understanding network-level energetics, an area of significant research interest[3,11,16,18,28].

We observed the highest and most variable contributions of EEMS regions III and V to the total FDOM pool prior to peak discharge, suggesting the landscape is hydrologically connected during snowmelt. These fluorescent regions are associated with aromatic, terrestrial-derived organics[29,30] and linked to DOM mobilization across the terrestrial-aquatic interface[5,31]. In sub-alpine watersheds, initial changes in river water chemistry are driven by snowmelt activation of shallow soil flowpaths[32]. Spring thaw rapidly increases landscape hydrologic connectivity, releasing metabolites that have accumulated in isolated soil pores throughout the winter[33]. Materials transported from near-surface soil compartments are enriched in reactive plant-derived compounds, especially phenolics and other lignin derivatives[34]. We observed high concentrations of metabolites with relatively lower parent ion *m/z* values, which support activation of shallow flowpaths during snowmelt and export of plant-derived materials[35], including syringol and isovannilic acid, which are associated with lignin-type derivatives[23]. Snowmelt water that intersects deeper soil flowpaths mobilizes compounds enriched in microbial-derivatives and nitrogen-bearing compounds[36]. Thus, river water chemistries exported in the spring may reflect activation of diverse soil flowpaths[31], however, rapid export during peak flows may result in mixing and convergence of the chemical profile. Although high concentrations of potentially labile DOM are exported with snowmelt, cold temperatures, rapid flow velocities, and relatively low solar insolation likely limit in-stream metabolism[5].

During peak flows, we observed strongly homogenized fluvial and metabolite chemistries, and hyporheic scouring in single-channel segments. We propose these simplified reaches typify the "pulse-shunt" concept proposed by Raymond et al.[37], where high concentrations of terrestrial-derived DOM mobilized during snowmelt can bypass local metabolism due to rapid increases in

**Table 3 Annotated metabolites collected from subalpine watersheds**

| Compound | Score | Ion intensity | m/z | Retention time (s) | Molecular weight (g/mol) | KEGG classification |
|---|---|---|---|---|---|---|
| Lyxose | 2 | 1.07E + 08 | 73.09 | 905.41 | 150.13 | Carbohydrate (pentose sugar) |
| Dodecaethylene glycol | 2 | 1.04E + 08 | 117.11 | 1,048.61 | 546.65 | Ether (polyethylene glycols) |
| 2-hydroxy-pyridine | 2 | 9.15E + 07 | 117.12 | 883.29 | 95.10 | Pyridinones |
| Glycerol | 2 | 2.89E + 08 | 131.12 | 842.33 | 92.09 | Sugar alcohol |
| Triethylene glycol | 2 | 5.48E + 08 | 144.14 | 157.86 | 150.17 | Ether (polyethylene glycols) |
| Galactose | 2 | 7.25E + 07 | 144.14 | 259.73 | 180.16 | Hexose (carbohydrate) |
| Tripropylene glycol mono-n-butyl ether | 2 | 2.02E + 07 | 151.08 | 360.78 | 248.36 | Ether (polyethylene glycols) |
| Syringic acid | 2 | 1.93E + 08 | 152.07 | 191.13 | 198.17 | Gallic acid and derivatives |
| Lumichrome | 2 | 2.37E + 08 | 187.16 | 755.85 | 242.23 | Flavin pigment |
| Palmitic acid | 2 | 5.26E + 06 | 189.09 | 344.35 | 256.42 | Long-chain fatty acid |
| Arabinofuranose | 2 | 3.57E + 07 | 217.12 | 457.45 | 150.13 | Pentose (carbohydrate) |
| Isovanillic acid | 2 | 3.20E + 06 | 219.14 | 848.21 | 168.15 | P-methoxybenzoic acids and derivatives |
| Heptaethylene glycol | 2 | 1.33E + 06 | 295.16 | 738.69 | 326.39 | Ether (polyethylene glycols) |
| Propylene glycol | 2 | 1.59E + 07 | 297.10 | 551.73 | 76.09 | Secondary alcohol |
| Benzoic acid | 2 | 5.88E + 06 | 317.16 | 563.13 | 122.12 | Benzoic acid |
| Coumaric acid | 2 | 3.02E + 06 | 361.17 | 1,033.60 | 164.05 | Hydroxycinnamic acids and derivatives |
| Sucrose | 2 | 2.79E + 07 | 371.29 | 848.38 | 342.30 | O-glycosyl compounds (disaccharides) |
| Dodecanoic acid | 2 | 1.36E + 06 | 455.29 | 1,008.71 | 200.32 | Medium-chain fatty acid (lipids) |
| Stearic acid | 2 | 1.00E + 06 | 473.23 | 994.22 | 284.48 | Long-chain fatty acid (lipids) |
| Glyceric acid | 2 | 1.15E + 06 | 502.20 | 1,018.32 | 106.08 | Sugar acids and derivatives |
| Nonanoic acid | 2 | 9.43E + 05 | 506.07 | 983.12 | 158.23 | Medium-chain fatty acid (lipids) |
| Thiazole | 2 | 1.07E + 06 | 521.79 | 943.04 | 85.13 | Thiazole |
| Decaethylene glycol | 2 | 6.69E + 05 | 589.26 | 1,024.46 | 458.55 | Ether (polyethylene glycols) |

The ion intensity and *m/z* values are reported for the largest ion fragment within each clustered spectrum (where each spectrum represents a single TMS-derived metabolite). Lower parent ion *m/z* values indicate higher compound volatility and are typically associated with faster elution. Spectra were clustered using RamClust. Annotation scores were acquired by querying NIST Standard Reference and KEGG databases, and classified using ClassyFire, a taxonomic database. InChI identifiers are reported in Supplementary Table 4
*TMS* trimethylsilyl, *KEGG* Kyoto Encyclopedia of Genes and Genomes

river velocity. Systems with more homogenous flows have also been linked with lower microbial biodiversity and DOM uptake relative to sites with longer residence times that facilitate opportunities for microbial metabolism[17]. The lack of upstream DOM processing in more simple, homogenous valley corridors has implications for headwater carbon storage and water quality as well as the redistribution of carbon and nutrient cycling to downstream, higher-order rivers.

Like many mountainous headwaters, the complex North Saint Vrain (NSV) network exhibits pronounced downstream variations in valley width and river planform[6] that moderate pulse-shunt dynamics even during high-flow periods. Specifically, single-thread segments alternate with broad valley segments containing multi-thread channel morphologies that can promote hydrological buffering by dissipating transport energy[9]. Wegener et al.[11] showed that the multi-thread beaver meadow complex exhibits variable sink-source dynamics as a function of flow. During high-flow periods, water and entrained nutrients are distributed laterally in floodplain surface water bodies and are gradually released during low-flow periods. The release of these components subsidizes downstream metabolism, integrating ecosystem metabolism with landscape connectivity[38].

Throughout the summer (June–August), we observed significant reductions in stream velocity and increasing indicators of autochthonous productivity, suggesting greater opportunities for in-stream metabolism and decreased lateral connectivity across the terrestrial-aquatic interface[5]. Increasing relative contributions of soluble microbial-type proteins and decomposition byproducts (fluorescence regional integration (FRI) regions I, II, and IV) to total FDOM were particularly pronounced in the beaver pond and the beaver meadow complex, which are less shaded by

riparian vegetation and mountains than more confined sites. In addition, these low-flow sites have increased DOM residence times, and are typically warmer[11], enhancing geophysical opportunities for metabolism relative to high-flow sites[3]. Within the beaver pond, we observed high concentrations of saturated medium-chain (9:0 and 10:0) and long-chain (16:0 and 18:0) fatty acids, both of which have been linked to fungal and bacterial detritivores[39], and can serve as significant energy sources for aquatic heterotrophs.

During late summer baseflow (September–October) we observed significant landscape fragmentation as side and main channels within the beaver meadow complex became hydrologically disconnected[9]. Resulting fluctuations in redox gradients have been linked to the activation of metabolic pathways utilizing alternate terminal electron acceptors to degrade DOM[2,40,41]. Together with greater light exposure and photo-oxidative transformation[42,43], these conditions could contribute to observed increases in chemical diversity with hydrograph recession. Alternatively, shifts in DOM composition could be explained by changes in hydrologic flow path, which transition from soil matrix to groundwater inputs during baseflow[5,44]. However, groundwater sources are typically associated with lower molecular weight, less chemically diverse DOM[31,34]. Thus, while the increased contribution of soluble proteins (FRI regions I, II, and IV) to total FDOM could be explained by decreased loading of matric porewater from upland sources (enriched in FRI Regions III and V), the overall increase in metabolite diversity suggests microbial metabolism and photo-oxidation partially regulates increases in DOM heterogeneity as flows subside.

While seasonally driven increases in photo-oxidation[2] could explain some of the divergence in DOM composition with

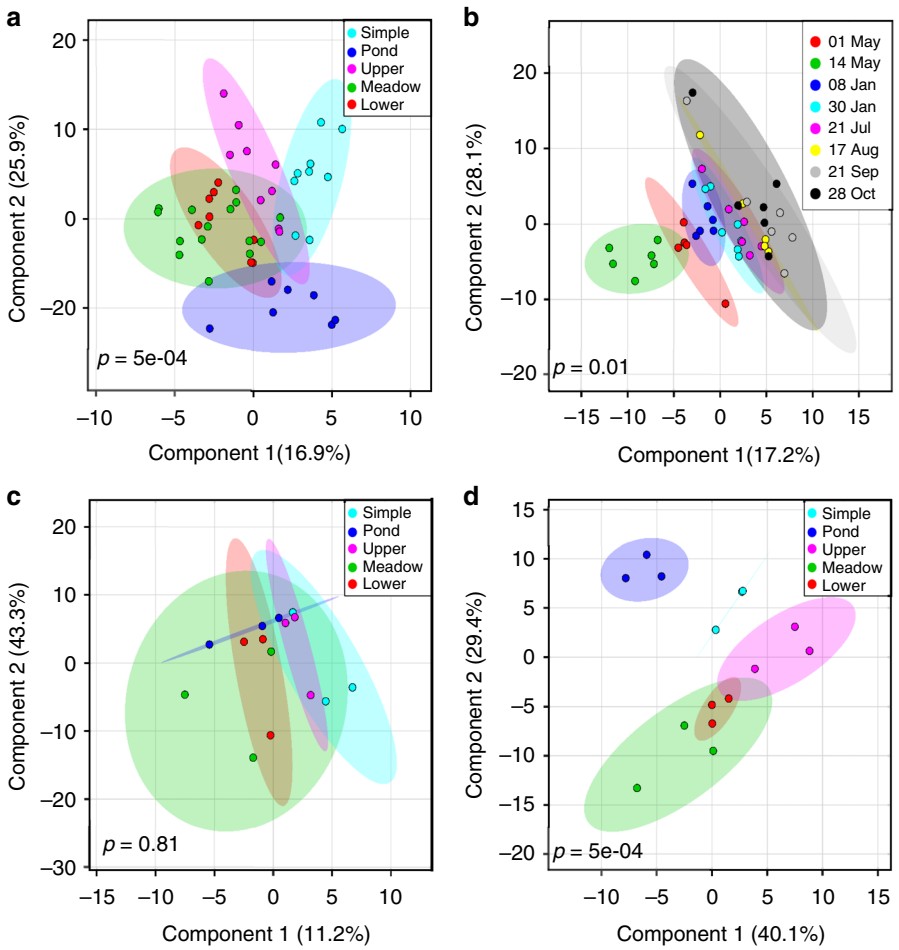

**Fig. 3** Partial least squares discriminant analysis (PLS-DA) plot of metabolites. Metabolites are clustered by landscape position (**a**), sampling period (**b**), rising limb, including only May 01, May 15, and June 08 (**c**), or baseflow, including only Aug 17, Sep 21, and Oct 28 (**d**). Each circle represents a sample and shaded ellipses represent 95% confidence intervals for each a priori cluster. Ellipse overlap signifies no significant difference between clusters. For each PLS-DA model, 2000 permutations were generated to test whether the classification systems (landscape position and sampling period) were significantly better ($p < 0.05$) than those arising through random chance

hydrograph recession, results presented here appear to support a shift from anabolic assimilation toward catabolic metabolism[19]. Specifically, we did not observe an increase in common photo-products, such as acetic, citric, pyruvic, or formic acids, which may have been rapidly metabolized[45,46]. Instead, declines in nutrient concentrations and hydrologic connectivity—assessed as a reduction in discharge—were associated with warmer temperatures in disconnected slough features and higher ecosystem respiration within the beaver meadow complex[11]. These patterns suggest nutrient-limited microbial communities cycle DOM less efficiently during warmer, low-flow periods, promoting carbon release to the atmosphere rather than incorporation in microbial biomass[47]. Unlike anabolism, which promotes chemical convergence through assimilation, catabolic respiration preserves the inherent complexity of DOM[19]. Thus, we revise our initial hypothesis and propose instead that carbon chemistry in sub-alpine rivers tends toward chemostasis during high-flow periods, when landscape connectivity and hydrologic drivers shunt DOM through the fluvial network, and hydrologic fragmentation shifts the balance from anabolic to catabolic metabolism, thereby increasing carbon complexity (Fig. 5).

Although still in its nascence, the ability to infer microbial functionality from metabolomics is a promising avenue of research[18,23]. EI GC-MS provides fragmentation spectra useful for identifying individual metabolites, but databases are sparse for environmental applications[48]. For example, products of second-ary metabolism (aromatics, alkaloids, terpenoids, glycosides, phenolics, and lignins) are not represented as well as those of primary metabolism (amino acids, sugars, organic acids, and peptides)[20]. We found the majority of compounds separating samples by landscape position had relatively high parent ion values m/z (>300), potentially indicative of secondary metabolites and other compounds with relatively high molecular weights.

The role of secondary metabolites in structuring network-level energetics is an area of considerable interest[35,49–51]. The environmental ubiquity of these structurally complex compounds suggests they confer a competitive advantage upon the organisms producing them. Otherwise, the pressures of Darwinian natural selection would preclude their synthesis[49]. Mounting evidence suggests secondary metabolites are variously involved in gene expression and cellular growth, with the potential to create more beneficial environmental conditions (for instance, by complexing iron and nutrients)[50]. Others have suggested the production of diverse metabolites facilitates microbial adaptation to complex environmental stressors[51], which are pronounced in subalpine watersheds, particularly under a changing climate. Although we could not annotate many of these compounds, by pairing meta-bolomics with fluorescence spectroscopy, we were able to link

microbial metabolism and landscape-scale processes to make inference about how underlying metabolic pathways structure local patterns of DOM chemistry throughout the fluvial network.

The ability of subalpine watersheds to store and process DOM is threatened by widespread and systematic channel simplification, resulting from land use changes, flow regulation, instream beaver and wood removal, and river channelization[11,52,53]. In addition to loss of geomorphic complexity, changes in hydrologic

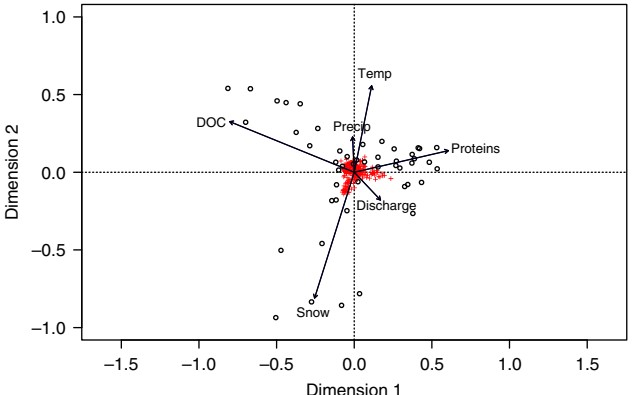

**Fig. 4** Multivariate analysis of metabolites and environmental drivers using capscale. Constrained analysis of principal coordinates (capscale) ordinations are based on Bray-Curtis dissimilarity, computed from normalized peak intensities, and used to test for an association of metabolite composition (as derived from electron ionization gas chromatography mass spectrometry) with spatial, climate, and fluvial characteristics. Red crosses represent individual metabolites, black circles represent samples, black vectors represent environmental variables with a significance level of <0.05 (p-values based on analysis of variance with 1000 permutations; the relationship between all potential predictor variables and metabolites is shown in Supplementary Figure 8). DOC, dissolved organic carbon; Precip, month-averaged precipitation; Temp, month-averaged temperature; Proteins, EEMS region I; Discharge, daily average; Snow, daily snow depth at Wild Basin

regime associated with reduced snowpack and stochasticity in the timing and magnitude of runoff have the potential to greatly alter the spatio-temporal dynamics of hydrologic connectivity at the terrestrial-aquatic interface. These watersheds are also experiencing dramatic shifts in biogeochemical function, including increased nitrogen deposition from intensified industrial and agricultural production[54], episodic acidification[55], and reductions in DOM and water subsidies from shrinking glaciers[28]. The combination of these factors reduces ecosystem function, with implications for human health[56], the value of water and storage-associated flow regimes[57], fate and transport of point and non-point source pollution[53], and storm water management[58]. Here we suggest subalpine systems are particularly sensitive to channel simplification because the loss of geomorphic complexity reduces metabolic opportunities for DOM processing, autochthonous productivity, hydrologic connectivity, and the maintenance of flow path and DOM diversity. We also suggest seasonal reductions in hydrologic connectivity, exacerbated by reductions in snowpack[59], will heighten catabolic metabolism by reducing nutrient availability in disconnected features. The propagation of these novel chemistries fuels downstream metabolism, tying local biogeochemical processing to network-level energetics.

## Methods

**Site description.** NSV and Beaver Brook are headwater rivers located in Rocky Mountain National Park, Colorado, USA. Both rivers have nearly identical climate regimes and underlying geology, consisting of a granitic core[60], but differ substantially in channel planform complexity. The approximately 4-km study reach along NSV exhibits pronounced downstream variations in valley geometry and channel complexity (Table 1). Channel planform along NSV alternates between single and multi-thread channel segments as a result of beaver activity[6]. The total drainage area of the NSV study reach is ~90 km² with an elevation of ~2560 m. Vegetation is typical of the Colorado Front Range, where aspen (*Populus tremuloides*) and willow stands (*Salix spp.*) dominate the riparian zone, and Engelmann spruce (*Picea engelmannii*), lodge pole pine (*Pinus contorta*), and subalpine fir (*Abies lasiocarpa*) colonize upland sites[11]. Approximately 3-km long, the Beaver Brook study site is herein referred to as Upper Beaver Meadow (UBM). This site was once a complex channel segment but has been abandoned by beavers since the early twentieth century, resulting in channel simplification[13]. As a result, a narrow (1–2 m wide), single-thread channel has incised (1–2 m) into legacy beaver meadow sediment[61]. Within the portion of Rocky Mountain National Park where these sites are located, mean annual temperature is 5 °C with a summer average of 14 °C,

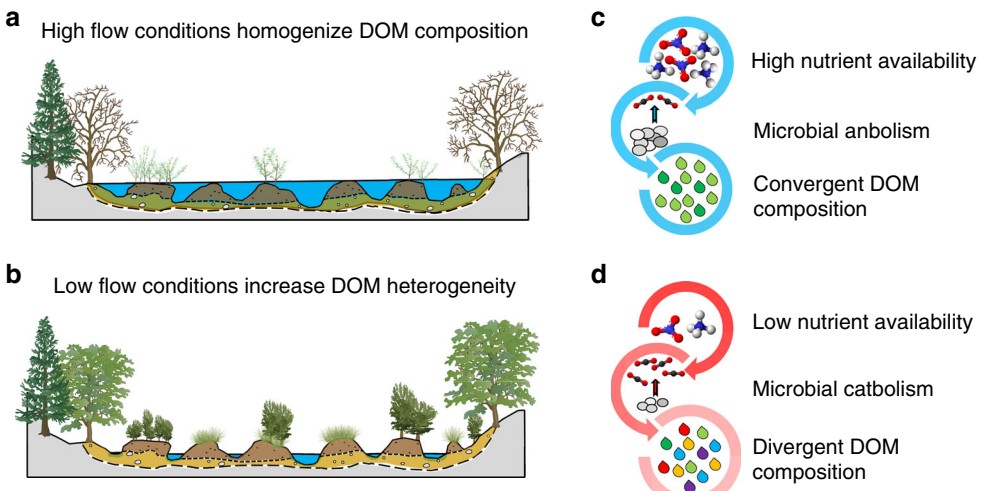

**Fig. 5** Conceptual model relating hydrologic connectivity and metabolite composition. During peak discharge (**a**), high flows limit geophysical opportunities for microbial metabolism. Landscape fragmentation during summer baseflow (**b**) increases water residence time and microbial processing of dissolved organic matter (DOM) and available nutrients. With sufficient nutrient availability, microbial anabolism dominates (**c**), causing DOM chemistry to converge toward a core set of metabolites. As nutrient availability decreases in isolated side-channel and slough features, catabolic pathways activate (**d**) and metabolite diversity increases. Seasonal changes in flow path, greater autochthonous productivity, and increased exposure to sunlight may also contribute to divergent DOM composition under low flow conditions. Ball and stick models depict carbon dioxide ($CO_2$), nitrate ($NO_3^-$), and ammonium ($NH_4^+$) molecules, where C atoms are shown in charcoal, O atoms in red, N atoms in blue, and H atoms in light gray

and mean annual precipitation is 861 mm, with 451 mm contributed as snowfall (Copeland Lake SNOTEL site # 412, 2621 m elevation, 40° 7′ 22.0794″N by 105° 20′ 26.5194″W).

Study reaches along UBM and NSV consist of three longitudinally adjacent river segments (Fig. 1; Table 1). At NSV we sampled a multi-thread, active beaver meadow complex bounded by relatively confined upstream and downstream single-thread segments. An upstream and downstream transect bounded each of the three NSV subreaches, and the NSV beaver meadow included an additional transect at the middle of the subreach. The NSV beaver meadow complex has the highest planform complexity, with multiple water impoundments behind beaver dams, as well as side channels and slough features that progressively disconnect from the main channel with hydrograph recession[11]. Samples were also collected from a beaver pond located along the upstream transect within the NSV beaver meadow, which has no surface water connectivity and sits ~3 m higher in elevation than the main channel. UBM is a relatively unconfined single channel segment with three sample locations separated by ~1500 m, and there are no multi-thread channels within the study reach.

The selection of these sites allowed us to assess the temporally dynamic role of lateral river-floodplain connectivity (high, low, and none) and longitudinal hydrologic connectivity (continuous, intermittent, and none) on fluvial biogeochemistry (Table 1). At each sampling site, we recorded GPS coordinates and physical characteristics including river (or pond) width and depth. River discharge was recorded hourly at UBM from June 11 through September 30, 2015 and at NSV from April 1 through August 30, 2015 (reported by Wegener et al.[11]).

**Sample collection and processing**. We collected paired surface water and hyporheic sediment samples eight times between May and September 2015 ($n = 48$, Supplementary Figure 1). Surface water samples were collected into sterile 50 mL amber borosilicate bottles and hyporheic sediments were collected into sterile 50 mL centrifuge vials. Additional large-volume surface water samples were collected into 10 L sterile containers and prepared for EI GC-MS analysis (see below). All samples were filtered within six hours of collection through 0.7 μm glass fiber filters pre-combusted at 400 °C (Whatman GF/F). Samples analyzed for total DOC and TDN were acidified to pH 3 and refrigerated until analysis on a Shimadzu TOC-L (Shimadzu Corporation Columbia, MD). Subsamples analyzed for inorganic nitrogen ($NO_3^-$ and $NH_4^+$) were frozen until analysis on an Alpkem flow solution IV automated wet chemistry system (O.I. Analytical College Station, TX).

**Excitation-emission matrices**. We used excitation-emission matrices to study the seasonal variability of UV-VIS FDOM characteristics[62]. To reduce inner-filter effects and normalize organic carbon concentrations, we diluted filtered samples to 5 mg C/L before analyzing them on an Aqualog spectrofluorometer with a xenon excitation source (Horiba-Jobin Yvone Scientific Edison, NJ). We set excitation and emission slits to a 3 nm band-pass, and incrementally increased wavelength in 3 nm steps from 200 to 800 nm. A sealed cuvette was used as a blank and analyzed between every twenty samples to correct for instrument drift. Following spectral analysis, we corrected each sample for inner-filter effects, masked Rayleigh scatter using first and second grating orders, and normalized each sample spectra by the blank[63]. We quantified percent intensities for regions of the EEMS using the FRI approach and separated five spectral regions as outlined by Chen et al.29 using Matlab version R2016b. EEMS regions I and II are related to simple aromatic proteins (tyrosine and tryptophan-like), region III to fulvic acid-type materials, region IV to soluble microbial byproduct-type materials, and region V to humic acid-type organics[29]. The FRI approach is better suited to capture the underlying heterogeneity of aromatic DOM as it quantifies entire regions of wavelength-dependent fluorescent intensities rather than utilizing only several data points per spectrum[29].

**Electron ionization gas chromatography-mass spectrometry**. We used EI GC-MS to generate mass spectral fragments, permitting partial structural identification of metabolites present in each sample[20]. We concentrated the total organic carbon in each pre-filtered 10 L surface water sample using four Bond Elut PPL cartridges pre-conditioned with high-performance liquid chromatography (HPLC)-grade methanol (Agilent Technologies). Following organic carbon concentration, each cartridge was extracted with 10 mL of HPLC-grade methanol into sterile borosilicate vials and evaporated under pure $N_2$ gas. Samples were prepared for GC-MS analysis at the Proteomics and Metabolomics Facility, Colorado State University. Samples were re-suspended in 5 mL of methanol and vortexed for 30 s. Next, 1 mL aliquots were collected and centrifuged for 10 min at 15,000 × g and 4 °C, and then 50 μL of N-methyl-N-trimethylsilyltrifluoroacetamide and 1% tri-methylchlorosilane were added to volatilize nonpolar metabolites. Sample supernatants were incubated for 30 min at 60 °C and then centrifuged for five minutes at 3000 × g. Next, 80 μL aliquots were transferred to a glass GC-MS autosampler vial and injected into a Trace GC Ultra coupled to a Thermo ISQ mass spectrometer (Thermo Scientific) in a 1:10 split ratio. A 30 m TG-5MS column (Thermo Scientific, 0.25 mm i.d., 0.25 μm film thickness) with a 1.2 mL/min helium gas flow rate separated metabolites. Masses between 50–650 mass to charge ($m/z$) were scanned at 5 scans/s after electron impact ionization. Quality control samples were injected after every six samples. A matrix of molecular features, defined by

retention time and $m/z$, were generated using XCMS software in R. We grouped all spectral features based on an in-house clustering tool, RAMClustR, which uses spectra based coelution and covariance across the full dataset to group features[21]. Peak areas for each spectral feature were condensed into a single value by adjusting the weighted mean of all spectral features for each compound. Compounds were annotated based on spectral matching to NISTv12, Golm, Metlin, Massbank, and in-house metabolite databases (Table 3)[21]. Two common contaminants, gallic acid and silanol, were also removed prior to subsequent analyses. We procured the InChI code for each annotated metabolite from PubChem and used ClassyFire to identify its chemical taxonomy[23] (Supplementary Table 4). Attempts were made to identify unknown metabolites contributing to sample separation in PLS-DA ordination space using Competitive Fragmentation Modeling for Metabolite Identification[24]. However, fragmentation patterns between the unknown and candidate metabolites were not robust; we include metabolite information and Jaccard scores in Supplementary Table 3 (confidence of assignment < 20% for all unknown compounds) and representative spectral similarity files in Supplementary Figure 9.

**Statistical analysis**. We used one-way analysis of variance (ANOVA) models to identify main effects of landscape position and collection period (Julian date) on each dependent variable of interest using the *lmtest* package[64] in R version 3.3.1[65]. We explored mean differences within each dependent variable using two-sample *t*-tests with unequal variances and Satterthwaite's approximation for degrees of freedom using the *lsmeans* package in R[66]. Data were log-transformed when necessary to meet assumptions of normality, evaluated with Shapiro-Wilk tests. Model residuals were tested for constant and homogenous variance using Q-Q and residual versus fitted plots. Outliers were identified using Cook's distance and removed if $D_i \geq 1$.

We used multiple linear regression analysis to identify relationships between our dependent variable of interest (DOC, EEMS region I, or EEMS region III) and potential predictor variables. Due to high correlations among many possible explanatory variables analyses (with $r \geq 0.6$), only those most consistent with past work or theory are included in our regression analyses (with $r \geq 0.6$). A correlation matrix of all possible predictors is included in Supplementary Figure 10. Potential variables for the initial regression analysis included channel width, river discharge, precipitation, temperature, valley confinement, and valley topographic aspect. Stepwise multiple linear regressions were conducted following Sutfin and Wohl[8], using the *step* function in the R *stats* package[65]. We normalized DOC using the boxcox power transformation in the R *MASS* package, with $\lambda = -0.343434$. Boxcox transformations yielded normally distributed errors, assessed with Shapiro-Wilk tests of the residuals (0.85), and inspection of Q-Q plots and histograms of model residuals.

We used MetaboAnalyst 3.0 for detailed metabolite processing and analysis[25]. We removed non-informative metabolites using interquartile range filtering, and normalized the data using a log transformation and Pareto variance scaling, which are common data pretreatment methods[67]. We identified significantly different metabolites using one-way ANOVA and Tukey's HSD post hoc analyses and determined whether samples clustered by categorical factors (landscape position or sampling period), using PLS-DA, a method commonly applied to metabolomic and chemometric datasets[68]. We performed PLS-DA regressions in MetaboAnalyst 3.0, which uses the *plsr* function in the R *pls* package[69]. We assessed the performance of each generated model using classification and 10-fold cross-validation and used prediction accuracy during training and B/W permutation tests to explore relationships between our metabolite distribution (predictor variables) and hypothetical classification system (landscape position or sampling period). For each PLS-DA model, we generated 2000 permutations to test whether these classification systems were significantly better than those arising through random chance[68]. We also used two unsupervised methods, PCA and random forest[70], to model the predictive capacity of our two classification systems.

We used constrained analysis of principal components (capscale), an ordination method in the R *vegan* package that permits non-Euclidean dissimilarity indices[27], to test for an association of metabolite composition (as derived from EI GC-MS) with spatial, climate, and fluvial characteristics. We included 25 potential environmental drivers in our ordination; a full capscale model displaying the association of all potential predictor variables with metabolite composition is provided in Supplementary Figure 8. We computed Bray-Curtis dissimilarities from normalized metabolite peak intensities and fit our environmental drivers to each ordination, with *p*-values calculated over 999 permutations. We extracted sample and metabolite information from the first two canonical axes to test for an association between metabolite characteristics ($m/z$, retention time, abundance), and retained ($p < 0.01$) environmental drivers. We used an independent non-metric multidimensional scaling ordination to validate retention of the selected environmental drivers (stress: 0.05).

## Data availability
All data used in this analysis are available through FigShare, an open-source data repository (https://doi.org/10.6084/m9.figshare.7015190).

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

## Acknowledgements

This research was supported by a National Science Foundation CAREER grant (number 255228) awarded to M.D.W., and a National Science Foundation IGERT fellowship (number 0966346) awarded to L.M.L. and N.A.S. We thank two anonymous reviewers for their thoughtful comments, which significantly improved our manuscript. We thank Dr. Ellen Wohl for assistance in selecting study sites, Pam Wegener for sharing discharge data, and Dr. Ann Hess for statistics consultation.

## Author contributions

L.M.L., N.A.S., and T.S.F. designed the study, with input from C.M.B., T.P.C., and M.D.W.; L.M.L., N.A.S., and T.S.F. conducted field work and analyzed the data; L.M.L. wrote the first draft of the manuscript; N.A.S., T.P.C., C.M.B., T.S.F., and M.D.W. commented and contributed to manuscript revisions.

## Additional information

**Competing interests:** The authors declare no competing interests.

