## [Transparent Peer Review File · Nature Communications]

Reviewers' comments:

Reviewer #1 (Remarks to the Author):

This paper discusses dissolved organic matter (DOM) analysis for hydrology and geomorphology research. The paper is written in a sound and technical manner. DOM is a highly complex matter and cannot be resolved by a single analytical technique. GC-MS is a feasible technique to obtain distinct fingerprints, as shown in this application. The processed result data and raw data are missing and should be submitted to the supplement and public repository.

Major issues:

1) The paper uses an older term "electron impact" which is obsolete: See IUPAC Gold Book.

Recommended is "electron ionization"

<https://goldbook.iupac.org/html/E/E01998.html>

<https://goldbook.iupac.org/html/E/E01999.html>

2) Add the full metabolite report used for statistical analysis as an EXCEL sheet to the supplement, add appropriate meta-data such as sample ID, location and metabolites and their IDs and annotations.

3) Add the raw data to a website or submit to <https://zenodo.org/> add the link to the supplement.

4) Regarding Table 3, while the unknown features maybe important, because they were derived from statistical analysis, they are of lesser interest to the reader because it is not known what they are. I would also include some of the known compounds and named compounds after the FDR correction. These are listed in the supplement but should go in the main manuscript.

If the unknowns are indeed important, because of PLS-DA variable importance, then efforts should be undertaken to annotate these. This can be done by adding some spectral similarity tables for each of the compounds to the supplement. This will allow the reader to obtain some compound class information. Another way would be to use software such as

<http://cfmid.wishartlab.com/identify>

5) Regarding the variable importance, perform an (multiclass) ANOVA and see if the important compounds are still the same. That is not always the case, because PLS-DA derives the variable importance by different algorithms. Nevertheless an F-test or in this case ANOVA should also reveal the important compounds. Also PLS-DA is prone to overfitting, especially with small sample sizes.

Minor issues:

1) The authors should shortly discuss in the introduction the rationale and advantage for selecting GC-MS, when classically Pyrolysis GC-MS or FT-MS was used for DOM analysis.

2) It would be interesting to actually see a picture of the TIC and the complexity of DOM. Add to supplement data.

4) I would recommend against the use of "C" for carbon throughout the text, but rather use carbon or organic carbon or similar terms. It makes the text more readable.

5) Line 655 "We concentrated the C" use "carbon" or "total organic carbon"

6) Line 656 "C concentration" same issue, use "organic matter" instead of "C"

7) PLS-DA is prone to overfitting; it would be good to add the (unsupervised) PCA plots to the supplement.

Reviewer #2 (Remarks to the Author):

This paper describes differences in dissolved organic matter (DOM) chemistry between sub alpine stream segments in Rocky Mountain National park in Colorado with contrasting landscape characteristics. These landscape characteristics are used to infer hydrologic connectivity and changes in DOM chemistry with season / hydrology are also described. The major claims of the

paper are that under high flow (snowmelt) periods subalpine headwater streams tend to export DOM downstream with little microbial alteration and that under lower flow conditions, stream segments with more geomorphic complexity exhibit a more heterogeneous pool of DOM when compared with simpler stream segments.

The first claim that DOM composition is defined by connectivity to the terrestrial landscape at high flow and that DOM is transported further downstream before alteration has been observed in a variety of headwater systems and led to the Pulse-Shunt concept (Raymond et al.) cited in the discussion section and other similar conceptual reviews. The second claim, that stream segment geomorphic complexity and landscape characteristics define the changes in DOM chemistry that occur with season / hydrology is more novel and the use of EI GC-MS to provide a more detailed characterization of DOM chemistry allows for further description of potential changes in source and microbial processing than more commonly used methods like 3d-EEMS.

•Will the paper be of interest to others in the field?

Yes, The paper be of interest to others in the field because the detailed characterization of DOM differences with hydrology/season by landscape interactions adds new insight into how sub-alpine aquatic ecosystems might be influenced by future climate and land management scenarios.

•Will the paper influence thinking in the field?

The broad suggestion that hydrology and landscape interact to shape molecular patterns of DOM is fairly well established, but this paper does provide important incremental advances in understanding the specifics of landscape factors influencing molecular patterns of DOM in sub-alpine systems given that previous work on this topic has largely focussed on other types of forested headwaters. I'm not well versed in mass spectrometry, but the additional detail provided by EI GC-MS indicates a number of potential avenues for new research on these topics.

•Are the claims convincing? If not, what further evidence is needed? Are the claims appropriately discussed in the context of previous literature?

For the most part this study is descriptive rather than experimental. For that reason I wonder if it might be possible to frame the paper with one or two broad research questions about the influence or geomorphology and seasonality/hydrology rather than with the hypotheses that note mechanisms not directly measured (lines 95-96). I'd rather see the hypotheses/ explanations for observed patterns presented in the discussion along with some of the alternative explanations. Some of this consideration of alternative explanations is lacking in the text of the discussion in its present form, but full consideration of alternative hypothesis is required for this type of study. It may be that the overall conclusions are altered very little, but some of the factors that influence DOM molecular pattern other than microbial processing require further consideration based on previous literature:

1) How does the potential for photochemical alteration of DOM differ between geomorphic or landscape settings in the study?

2) The focus of the authors in explaining the observed patterns (e.g. Fig 5) seems primarily focussed on the impact of heterotrophic processing of DOM originating from terrestrial sources, but to what extent might autotrophic production within the aquatic systems be important? There is some mention about the potential on line 164, but I would assume it may also impact the GC-MS results.

3) Although discharge tends to change seasonally, there are other factors that change with season including temperature (as noted in the manuscript), timing of senescence of vegetation, and soil moisture. Is it possible to differentiate between discharge driven seasonality and the influence of other drivers? Plotting a time series of sample timing and potentially some of the DOM chemistry on a hydrograph could be quite helpful for this.

4) What type of molecular signature is anticipated with an increasing proportion of flow occurring with groundwater input?

5) The overall suggestion that channel simplification alters DOM through reduced opportunities for

DOM processing is fine, but potential changes in flow path, autochthonous production, and photochemistry also need to be considered.

- If the manuscript is unacceptable in its present form, does the study seem sufficiently promising that the authors should be encouraged to consider a resubmission in the future?

There is a tendency in the text of the manuscript to focus on preferred explanations of the observed seasonal and spatial patterns of DOM chemistry. The description of the patterns themselves are quite interesting and the methods used are sound, so the authors should be encouraged to make some revisions to the manuscript so that alternative explanations are more clearly addressed (as suggested above).

- Is the manuscript clearly written? If not, how could it be made more accessible?

The manuscript is well written, but there were a few places where sentences were tough to read because of length or where the wording needs to be more precise. These are noted at the end of this review.

- Should the authors be asked to provide further data or methodological information to help others replicate their work? (Such data might include source code for modelling studies, detailed protocols or mathematical derivations).

Hydrological data would be quite useful. Also, the inclusion of only “uncorrelated environmental variables” is a bit confusing. In the ordination for example why not include all environmental variables since the ordination space is only defined by the chemistry. For other analysis, a clearer statement of what was excluded and the selection criteria for inclusion is needed.

- Statistical analyses

I'm more familiar with PLS than PLS-DA, but to what extent is the PLS-DA space defined by the groupings selected? It would be useful to provide some speculation about what accounts for unexplained variation and whether selection of a different set of predictors or categories might change the interpretation of results. For example, what would happen if sites were grouped by surface area or a categorical version of any of the other continuous variables shown in Figure 4?

Specific comments:

Line 34 – “dominant” in what way?

Line 44 - Are there other forms of C retention in-stream?

Line 47 – The connection of channel geometry and planform to residence time are quite clear, but further explanation / referencing on the influence on concentration of DOM seems to be needed here.

Line 129 – This sentence is challenging to interpret. I assume this means average of the 3 UBM sampling locations for each point in time.

Line 140 – “export” ... should this read concentration, since export is almost always higher with peak flow

Line 157 – I don't see fluorescence information in Table 2.

Line 164 – Since these are relative % intensities couldn't higher proportion of autochthonous signal occur without increase in productivity, but just a decrease in loading from upland sources? It may also be worth noting in discussion rather than the results that this signal also has been observed in groundwater, so flow path might also be having an influence.

Line 302 – “non-targeted metabolomics” – a definition in the text would be helpful. It's not a term I have come across frequently in the stream ecology or biogeochemistry literature.

Line 309- What does it mean for the landscape to be “integrate”?

Lines 315-324 – This interpretation of the dataset is really interesting and adds detail to findings using other methods of molecular characterization.

Line 345- Hydrological connectivity wasn't directly measured was it? Is this indicated by high or low flow?

Line 346 – “Availability” is relative, so maybe concentration would be a better term here.

Lines 344-350 – The potential for changes in autochthonous production and contributions to

microbial assimilation and catabolic metabolism needs to be considered here.

Figure 5- Maybe photoprocessing, flow path change, and primary production need to be considered as well

Line 414-416 – Should reduction in potential for autochthonous production and complexity of sources / flow paths with loss of geomorphic complexity also be mentioned here?

October 15, 2018
Nature Communications
Ms. Ref. No.: NCOMMS-18-14160A

Dear reviewers,

We would like to thank both of you very much for your constructive review of our manuscript, entitled 'River channel connectivity shifts metabolite composition and dissolved organic matter chemistry'. Throughout the document, our responses to each comment are italicized and shown in blue text. We believe your thoughtful suggestions have significantly improved our manuscript.

Best wishes,
Laurel Lynch

Reviewer #1 (Remarks to the Author):

This paper discusses dissolved organic matter (DOM) analysis for hydrology and geomorphology research. The paper is written in a sound and technical manner. DOM is a highly complex matter and cannot be resolved by a single analytical technique. GC-MS is a feasible technique to obtain distinct fingerprints, as shown in this application. The processed result data and raw data are missing and should be submitted to the supplement and public repository.

Major issues:

1) The paper uses an older term “electron impact” which is obsolete: See IUPAC Gold Book. Recommended is “electron ionization”

<https://goldbook.iupac.org/html/E/E01998.html>

<https://goldbook.iupac.org/html/E/E01999.html>

We updated ‘impact’ to ‘ionization’ throughout the text.

2) Add the full metabolite report used for statistical analysis as an EXCEL sheet to the supplement, add appropriate meta-data such as sample ID, location and metabolites and their IDs and annotations.

We have uploaded our entire dataset (including all spatial, environmental, and metabolite data used our statistical analyses) to FigShare. The DOI accession number is 10.6084/m9.figshare.7015190. The first tab provides our metadata, including sample ID, landscape and sampling period ID, and metabolite annotations. The second tab provides the spatial, environmental, and metabolite data used for all statistical analyses. The third tab provides the full metabolite report. The fourth tab provides the clustered spectrum of ion fragments for each TMS-derived metabolite and reports the absolute ion intensity and the parent ion m/z values.

3) Add the raw data to a website or submit to <https://zenodo.org/> add the link to the supplement.

The raw data are available as a .MSP library in the FigsShare data repository. The DOI access number is: 10.6084/m9.figshare.7015190.

4) Regarding Table 3, while the unknown features maybe important, because they were derived from statistical analysis, they are of lesser interest to the reader because it is not known what they are. I would also include some of the known compounds and named compounds after the FDR correction. These are listed in the supplement but should go in the main manuscript. If the unknowns are indeed important, because of PLS-DA variable importance, then efforts should be undertaken to annotate these. This can be done by adding some spectral similarity tables for each of the compounds to the supplement. This will allow the reader to obtain some compound class information. Another way would be to use software such as <http://cfmid.wishartlab.com/identify>

We have moved the complete list of annotated compounds to the main text (Table 4) and included spectral scores in the table. The InChiKey identifiers are provided in Supplementary Table 3 (VIP compounds with putative assignments) and Supplementary Table 4 (annotated metabolites).

As suggested, we used CFM-ID (Wishart Lab) to annotate previously unidentified VIP-selected variables (at the molecular or class-level). However, the quality of the spectral matches were low, with Jaccard scores ranging between 0.05 and 0.20. Because of these low scores (<20% similarity), we do not feel comfortable including this table in the main text, but have included parent ion m/z, intensity, and retention time information, as well as F-values and Jaccard scores reference (Supplementary Table 3). We have also included two representative spectral similarity figures (Supplementary Figure 8) for reference. metabolite data have been uploaded to FigShare for public access.

To the methods section we have added the following text:

“Attempts were made to identify unknown metabolites contributing to sample separation in PLS-DA ordination space using Competitive Fragmentation Modeling for Metabolite Identification (CFM-ID)²⁴. However, fragmentation patterns between the unknown and candidate metabolites were not robust; we include metabolite information and Jaccard scores in Supplementary Table 3 (confidence of assignment < 20% for all compounds) and representative spectral similarity files in Supplementary Figure 9.”

5) Regarding the variable importance, perform an (multiclass) ANOVA and see if the important compounds are still the same. That is not always the case, because PLS-DA derives the variable importance by different algorithms. Nevertheless, an F-test or in this case ANOVA should also reveal the important compounds. Also, PLS-DA is prone to overfitting, especially with small sample sizes.

Thank you very much for this suggestion. We included the F-statistics and p-values for each PLS-DA selected metabolite. With the exception of one compound each (across landscape position and sampling period) the same metabolites were selected, although the ordering was slightly different between PLS-DA and ANOVA models.

To address issues of model overfitting, we have included Q^2 values for each PLS-DA model in the main text (lines 157 and 184). The Q^2 values for each model are positive and relatively large, suggesting neither model is overfitted. However, we have included two additional statistical approaches in the supplementary information to support our interpretation. The first, as suggested, is an unsupervised PCA plot (Supplementary Figure 5). The second is output from random forest modeling, using 5,000 trees (Supplementary Figure 6). Random forest classification uses a stochastic, ensemble approach; during tree construction, one-third of the samples are not included in the bootstrap sample and are used as test data to obtain unbiased estimates of the (out-of-bag) classification error. Both PCA and random forest results support our main conclusions from the PLS-DA analysis: during high flow periods, similar metabolite profiles are exported regardless of landscape position; as flows subside, variability in metabolite diversity increases between landscape positions.

Minor issues:

1) The authors should shortly discuss in the introduction the rationale and advantage for selecting GC-MS, when classically Pyrolysis GC-MS or FT-MS was used for DOM analysis.

We have updated the introduction as follows: “Mass spectrometry has vastly improved our understanding of the composition and cycling of DOM compounds varying in physicochemical reactivity. While fluorescence, NMR, pyrolysis GC-MS, and FT-ICR-MS approaches are routinely used to probe DOM composition, they group DOM complexity into broad functional groups or compound classes. In contrast, electron ionization gas chromatography mass spectrometry (EI GC-MS) can identify individual metabolites from complex mixtures of environmental DOM²⁰. This detailed characterization of DOM allows inferences about the underlying patterns of microbial metabolism that structure the reactivity and fate of DOM flowing throughout the fluvial network”.

In the discussion we comment upon the current state of EI GC-MS for ecological purposes: “Although still in its nascence, the ability to infer microbial functionality from metabolomics is a promising avenue of research. EI GC-MS provides fragmentation spectra useful for identifying individual metabolites, but databases are sparse for ecological purposes. For example, products of secondary metabolism (aromatics, alkaloids, terpenoids, glycosides, phenolics, lignins) are not represented as well as those of primary metabolism (amino acids, sugars, organic acids, peptides)”.

2) It would be interesting to actually see a picture of the TIC and the complexity of DOM. Add to supplement data.

We have added representative total ion chromatograms for each landscape position during peak discharge or baseflow conditions (Supplementary Figure 8).

4) I would recommend against the use of “C” for carbon throughout the text, but rather use carbon or organic carbon or similar terms. It makes the text more readable.

We changed ‘C’ to ‘carbon’ throughout the text to improve readability. We also changed ‘N’ to ‘nitrogen’ throughout the text.

5) Line 655 “We concentrated the C” use “carbon” or “total organic carbon”

We changed the text to ‘total organic carbon’.

6) Line 656 “C concentration” same issue, use “organic matter” instead of “C”

We changed the text to ‘organic carbon concentration’.

7) PLS-DA is prone to overfitting; it would be good to add the (unsupervised) PCA plots to the supplement.

Unsupervised PCA plots are supplied as Supplementary Figure 5 and referred to in the caption of Figure 3 and main text. We also include results using a random forest approach (Supplementary Figure 6), which eliminates overfitting. Both the PCA and random forest model output provide additional means of assessing how the compositional complexity of DOM varies seasonally and across landscape position.

The text now reads: “While our strongly positive score ($Q^2=0.43$) suggests the model is not overfitted²⁶ we also include results from two unsupervised approaches (principal components analysis, Supplementary Figure 5, and random forest, Supplementary Figure 6) that display similar trends.”

Reviewer #2 (Remarks to the Author):

This paper describes differences in dissolved organic matter (DOM) chemistry between sub alpine stream segments in Rocky Mountain National park in Colorado with contrasting landscape characteristics. These landscape characteristics are used to infer hydrologic connectivity and changes in DOM chemistry with season / hydrology are also described. The major claims of the paper are that under high flow (snowmelt) periods subalpine headwater streams tend to export DOM downstream with little microbial alteration and that under lower flow conditions, stream segments with more geomorphic complexity exhibit a more heterogeneous pool of DOM when compared with simpler stream segments.

The first claim that DOM composition is defined by connectivity to the terrestrial landscape at high flow and that DOM is transported further downstream before alteration has been observed in a variety of headwater systems and led to the Pulse-Shunt concept (Raymond et al.) cited in the discussion section and other similar conceptual reviews. The second claim, that stream segment geomorphic complexity and landscape characteristics define the changes in DOM chemistry that occur with season / hydrology is more novel and the use of EI GC-MS to provide a more detailed characterization of DOM chemistry allows for further description of potential changes in source and microbial processing than more commonly used methods like 3d-EEMS.

•Will the paper be of interest to others in the field?

Yes, the paper be of interest to others in the field because the detailed characterization of DOM differences with hydrology/season by landscape interactions adds new insight into how sub-alpine aquatic ecosystems might be influenced by future climate and land management scenarios.

•Will the paper influence thinking in the field?

The broad suggestion that hydrology and landscape interact to shape molecular patterns of DOM is fairly well established, but this paper does provide important incremental advances in understanding the specifics of landscape factors influencing molecular patterns of DOM in sub-alpine systems given that previous work on this topic has largely focused on other types of forested headwaters. I'm not well versed in mass spectrometry, but the additional detail provided by EI GC-MS indicates a number of potential avenues for new research on these topics.

•Are the claims convincing? If not, what further evidence is needed? Are the claims appropriately discussed in the context of previous literature?

For the most part this study is descriptive rather than experimental. For that reason, I wonder if it might be possible to frame the paper with one or two broad research questions about the influence or geomorphology and seasonality/hydrology rather than with the hypotheses that note mechanisms not directly measured (lines 95-96).

Thank you for this suggestion. We have framed the paper with two broad research questions and have modified our discussion to include the suggested mechanisms, which complement our interpretation (including, photo-oxidation, groundwater inputs, and seasonal by landscape

position interactions). We have framed our paper with two broad hypotheses: “We posit that (1) river segments with high geomorphic complexity increase the chemical diversity of DOM exported downstream, and (2) maximal chemical diversity occurs during high flow periods and declines over time as flows subside.”

I'd rather see the hypotheses/ explanations for observed patterns presented in the discussion along with some of the alternative explanations. Some of this consideration of alternative explanations is lacking in the text of the discussion in its present form, but full consideration of alternative hypothesis is required for this type of study. It may be that the overall conclusions are altered very little, but some of the factors that influence DOM molecular pattern other than microbial processing require further consideration based on previous literature:

We have extensively updated the text and our conceptual model based on these recommendations (individual responses provided below).

1) How does the potential for photochemical alteration of DOM differ between geomorphic or landscape settings in the study?

We have extensively edited the text to include explicit discussions of photo-oxidation (including below). We added the following text to the discussion that highlights how the potential for photo-oxidative alteration of DOM chemistry may vary by landscape position:

“Throughout the summer (June – August), we observed significant reductions in stream velocity and increasing indicators of autochthonous productivity, suggesting greater opportunities for in-stream metabolism and decreased interaction across the terrestrial aquatic interface⁵. Increasing relative contributions of soluble microbial-type proteins and decomposition byproducts (FRI regions I, II, and IV) to total FDOM were particularly pronounced in the beaver pond and the beaver meadow complex, which are less shaded by riparian vegetation and mountains than more confined sites. In addition, these low-flow sites have increased DOM residence times, and are typically warmer¹¹, enhancing geophysical opportunities for metabolism relative to high-flow sites³. Within the beaver pond, we observed high concentrations of saturated medium-chain (9:0 and 10:0) and long-chain (16:0 and 18:0) fatty acids, both of which have been linked to fungal and bacterial detritivores³⁹, and can serve as significant energy sources for aquatic heterotrophs.

During late summer baseflow (September – October) we observed significant landscape fragmentation as side and main channels within the beaver meadow complex became hydrologically disconnected⁹. Resulting fluctuations in redox gradients have been linked to the activation of metabolic pathways utilizing alternate terminal electron acceptors to degrade DOM^{2,40,41}. Together with greater light exposure and photo-oxidative transformation^{42,43}, these conditions could contribute to observed increases in chemical diversity with hydrograph recession. Alternatively, shifts in DOM composition could be explained by changes in hydrologic flowpath, which transition from soil matrix to groundwater inputs during baseflow^{5,44}. However, groundwater sources are typically associated with lower molecular weight, less chemically diverse DOM^{31,34}. Thus, while the increase in the contribution soluble proteins (FRI Regions I, II, and IV) to total FDOM could be explained by decreased loading of matric porewater from upland sources (enriched in FRI Regions III and V), the overall increase in metabolite diversity

suggests microbial metabolism and photo-oxidation partially regulates increases in DOM heterogeneity as flows subside.

While seasonally driven increases in photo-oxidation² could explain some of the divergence in DOM composition with hydrograph recession, results presented here appear to support a shift from anabolic assimilation toward catabolic metabolism¹⁹. Specifically, we did not observe an increase in common photoproducts, such as acetic, formic, citric, pyruvic, or formic acids, which may have been rapidly metabolized^{45,46}. Instead, declines in nutrient concentrations and hydrologic connectivity—assessed as a reduction in discharge—were associated with warmer temperatures in disconnected slough features and higher ecosystem respiration within the beaver meadow complex¹¹.”

2) The focus of the authors in explaining the observed patterns (e.g. Fig 5) seems primarily focused on the impact of heterotrophic processing of DOM originating from terrestrial sources, but to what extent might autotrophic production within the aquatic systems be important? There is some mention about the potential on line 164, but I would assume it may also impact the GC-MS results.

We added a paragraph (lines 259-281) that weaves autotrophic production in subalpine systems into the main discussion (see above). We have also significantly updated figure 5 to highlight shifts in flowpath, autotrophic potential, and the influence of vegetation (specifically leaf-out) on DOM composition.

3) Although discharge tends to change seasonally, there are other factors that change with season including temperature (as noted in the manuscript), timing of senescence of vegetation, and soil moisture. Is it possible to differentiate between discharge driven seasonality and the influence of other drivers? Plotting a time series of sample timing and potentially some of the DOM chemistry on a hydrograph could be quite helpful for this.

We have added a hydrograph, with sampling period indicated by vertical bars (Supplementary Figure 1). We have also included representative total ion chromatograms for each landscape position during peak discharge or baseflow conditions (Supplementary Figure 3).

4) What type of molecular signature is anticipated with an increasing proportion of flow occurring with groundwater input?

This is an intriguing question, and the focus of current work attempting to identify the controls on DOM signature and reactivity at watershed scales. We expect that during snowmelt, substantial volumes of ‘young’ water are exported downstream (i.e. Kirchner’s ‘A double paradox in catchment hydrology and geochemistry’). During baseflow, older, percolated water, is exported. Work by Burns et al. at similar sites suggests groundwater sources contribute lower molecular weight, less chemically diverse DOM to rivers. We have included a discussion of baseflow chemistry in the text as follows: “Together with greater light exposure and photo-oxidative transformation^{42,43}, these conditions could contribute to observed increases in chemical diversity with hydrograph recession. Alternatively, shifts in DOM composition could be explained by changes in hydrologic flowpath, which transition from soil matrix to groundwater inputs during baseflow^{5,44}. However, groundwater sources are typically associated with lower

molecular weight, less chemically diverse DOM^{31,34}. Thus, while the increased contribution of soluble proteins (FRI Regions I, II, and IV) to total FDOM could be explained by decreased loading of matrix porewater from upland sources (enriched in FRI Regions III and V), the overall increase in metabolite diversity suggests microbial metabolism and photo-oxidation partially regulates increases in DOM heterogeneity as flows subside.”

5) The overall suggestion that channel simplification alters DOM through reduced opportunities for DOM processing is fine, but potential changes in flow path, autochthonous production, and photochemistry also need to be considered.

We have added additional depth to our discussion, exploring alternative explanations for our observed shifts in DOM compositional diversity.

•If the manuscript is unacceptable in its present form, does the study seem sufficiently promising that the authors should be encouraged to consider a resubmission in the future?

There is a tendency in the text of the manuscript to focus on preferred explanations of the observed seasonal and spatial patterns of DOM chemistry. The description of the patterns themselves are quite interesting and the methods used are sound, so the authors should be encouraged to make some revisions to the manuscript so that alternative explanations are more clearly addressed (as suggested above).

Thank you for this suggestion. We have incorporated the suggested, complementary mechanisms throughout the text, which has significantly improved the quality of our manuscript.

•Is the manuscript clearly written? If not, how could it be made more accessible?

The manuscript is well written, but there were a few places where sentences were tough to read because of length or where the wording needs to be more precise. These are noted at the end of this review.

Thank you. We have revised the text as recommended (see below).

•Should the authors be asked to provide further data or methodological information to help others replicate their work? (Such data might include source code for modelling studies, detailed protocols or mathematical derivations).

Hydrological data would be quite useful. Also, the inclusion of only “uncorrelated environmental variables” is a bit confusing. In the ordination for example why not include all environmental variables since the ordination space is only defined by the chemistry. For other analysis, a clearer statement of what was excluded and the selection criteria for inclusion is needed.

We uploaded hydrological data to FigShare, an open-source data repository (DOI accession number: 10.6084/m9.figshare.7015190). In addition to mean discharge at the time of sampling, we included three additional hydrologic variables that capture nuances by considering flow conditions over each four-day period prior to and including the day of sampling. These integrated variables (mean discharge, variance in discharge, and the slope of the best-fit line) capture diel variability, potential for hydrologic responses to recent storms, and the seasonality of flow conditions.

We have amended the caption for Figure 4, which should say ‘statistically significant’ rather than ‘uncorrelated’. The caption now reads: ‘Blue vectors represent variables with a significance level $p < 0.05$ (p-values based on ANOVA with 1,000 permutations; the relationship between all potential predictor variables and metabolites is shown in Supplementary Figure 8).’ The supplementary figure shows the relationship between all potential environmental variables and metabolites, which clarifies the variables included or excluded in each step of our multivariate analysis. Figure 4 (main text) now indicates that only statistically significant variables are retained in the supplied ordination plot.

•Statistical analyses

I’m more familiar with PLS than PLS-DA, but to what extent is the PLS-DA space defined by the groupings selected? It would be useful to provide some speculation about what accounts for unexplained variation and whether selection of a different set of predictors or categories might change the interpretation of results. For example, what would happen if sites were grouped by surface area or a categorical version of any of the other continuous variables shown in Figure 4?

PLS-DA space can be defined by the selected groupings or categorize groups independently of our a priori classifications. Our results are similar when plotted either way. We have included Q^2 scores in the main text of our paper. Both of our scores are strongly positive, indicating the models are not overfitted. However, we have included two additional statistical modeling approaches that are not influenced by a priori classification systems as Supplementary Figure 5 (PCA) and Supplementary Figure 6 (Random Forest).

We selected our two categorical groups a priori (landscape position and sampling period) because they capture shifts across multiple continuous and/or numeric variables (i.e. elevation, channel confinement, discharge, temperature). Because discharge varies seasonally and across landscape position, it is not a good grouping factor (each sample would have a unique value) and we wanted to avoid arbitrary binning. The predictive capacity of the models may be improved by including metrics not explicitly tested in this paper, including microbial community diversity (genomics), the physical heterogeneity of biofilms as a function of season, and the mean residence time of water at each landscape position.

Specific comments:

Line 34 – “dominant” in what way?

We changed this to: “Dissolved organic matter (DOM) accounts for the vast majority of reactive carbon flowing through aquatic ecosystems” to distinguish between bioavailable and buried DOM or particulate organic matter resources.

Line 44 - Are there other forms of C retention in-stream?

We changed this to: “By some estimates, half of this carbon is mineralized through microbial and photochemical degradation, while the fraction remaining within the watershed is retained

on floodplains, exported further downstream, or generated in situ through autochthonous production or heterotrophic fractionation.”

Line 47 – The connection of channel geometry and planform to residence time are quite clear, but further explanation / referencing on the influence on concentration of DOM seems to be needed here.

We added three references to this statement.

Line 129 – This sentence is challenging to interpret. I assume this means average of the 3 UBM sampling locations for each point in time.

We changed this sentence to read: “Therefore, for each sampling point we used an average of the three sampling locations along the simple network (UBM) for subsequent statistical analyses.”.

Line 140 – “export” ... should this read concentration, since export is almost always higher with peak flow

We corrected this sentence as follows: Regressing TDN against discharge revealed N concentrations were highest during peak river flows ($R^2=0.46$).

Line 157 – I don’t see fluorescence information in Table 2.

Fluorescence information was added as Supplementary Table 2, and figure/table numbering was updated throughout the text.

Line 164 – Since these are relative % intensities couldn’t higher proportion of autochthonous signal occur without increase in productivity, but just a decrease in loading from upland sources? It may also be worth noting in discussion rather than the results that this signal also has been observed in groundwater, so flow path might also be having an influence.

This is a great point and could definitely be contributing to shifts in DOM composition over time. However, we also observed an increase in metabolite diversity, which is independent of the relative % intensities. We have highlighted this discussion in the text: “Alternatively, shifts in DOM composition could be explained by changes in hydrologic flowpath, which transition from soil matrix to groundwater inputs during baseflow. However, groundwater sources are typically associated with lower molecular weight, less chemically diverse DOM. Thus, while the increase in the contribution of soluble proteins (FRI Regions I, II, and IV) to total FDOM could be explained by decreased loading of matric porewater from upland sources (enriched in FRI Regions III and V), the overall increase in metabolite diversity suggests microbial metabolism and photo-oxidation increase DOM heterogeneity as flows subside.”

Line 302 – “non-targeted metabolomics” – a definition in the text would be helpful. It’s not a term I have come across frequently in the stream ecology or biogeochemistry literature.

We define non-targeted as follows: While these bulk chemical approaches revealed the influence of seasonality, we found the application of non-targeted metabolomics—a higher-resolution approach capable of profiling both known and unknown metabolites—was essential for resolving the influence of landscape complexity on DOM chemistry.

Line 309- What does it mean for the landscape to be “integrate”?

We replaced ‘integrated’ with ‘hydrologically connected’ for clarity.

Lines 315-324 – This interpretation of the dataset is really interesting and adds detail to findings using other methods of molecular characterization.

Thank you.

Line 345- Hydrological connectivity wasn’t directly measured was it? Is this indicated by high or low flow?

We changed this passage to read: “As hydrologic connectivity decreased—as assessed by reductions in discharge—we observed...”

Line 346 – “Availability” is relative, so maybe concentration would be a better term here.

We changed ‘availability’ to ‘concentrations’.

Lines 344-350 – The potential for changes in autochthonous production and contributions to microbial assimilation and catabolic metabolism needs to be considered here.

We amended this section as follows: “In addition to greater autochthonous productivity, these patterns may also suggest nutrient-limited microbial communities cycle DOM less efficiently during warmer, low-flow periods, promoting carbon release to the atmosphere rather than incorporation in microbial biomass”

Figure 5- Maybe photo-processing, flow path change, and primary production need to be considered as well

We added the following text to the Figure 5 caption: “Seasonal changes in flow path, greater autochthonous productivity, and increased exposure to sunlight may also contribute to divergent DOM composition under low flow conditions.” We also updated the figure to more explicitly display these alternate mechanisms.

Line 414-416 – Should reduction in potential for autochthonous production and complexity of sources / flow paths with loss of geomorphic complexity also be mentioned here?

We have updated the text as follows: “Here, we suggest subalpine systems are particularly sensitive to channel simplification because the loss of geomorphic complexity reduces metabolic opportunities for DOM processing, autochthonous productivity, hydrologic connectivity, and the

maintenance of flowpath and DOM diversity.”

REVIEWERS' COMMENTS:

Reviewer #1 (Remarks to the Author):

I think most of the issues we raised were addressed or corrected.

Reviewer #2 (Remarks to the Author):

After revision, this paper remains well written and an interesting analysis of the relationship between landscape complexity, DOM complexity, and changes in this relationship with season. Through interpretation of detailed characterization of DOM chemistry, a clear set of hypotheses regarding a shift from anabolic assimilation toward catabolic metabolism of DOM with changing flow conditions is defined. The impact of landscape complexity in defining this response is also defined.

The authors have thoughtfully addressed reviewers concerns. My primary concern with the original manuscript was the need for increased consideration of flowpath (source)and light availability changes that accompany season and landscape differences. Consideration of environmental factors that may interact with microbial metabolic processes to shape DOM composition has been expanded in the revision.

We would like to thank you both again for your constructive review of our manuscript.

REVIEWER RESPONSES

Reviewer #1 (Remarks to the Author):

I think most of the issues we raised were addressed or corrected.

RESPONSE: Thank you for helping us clarify our statistical approaches and contextualize our EI GC-MS findings.

Reviewer #2 (Remarks to the Author):

After revision, this paper remains well written and an interesting analysis of the relationship between landscape complexity, DOM complexity, and changes in this relationship with season. Through interpretation of detailed characterization of DOM chemistry, a clear set of hypotheses regarding a shift from anabolic assimilation toward catabolic metabolism of DOM with changing flow conditions is defined. The impact of landscape complexity in defining this response is also defined.

The authors have thoughtfully addressed reviewers' concerns. My primary concern with the original manuscript was the need for increased consideration of flowpath (source) and light availability changes that accompany season and landscape differences. Consideration of environmental factors that may interact with microbial metabolic processes to shape DOM composition has been expanded in the revision.

RESPONSE: Thank you very much for your feedback. Expanding our discussion to include additional environmental drivers that may interactively shape DOM composition in subalpine watersheds has greatly improved our manuscript.